# Experimental Investigation of the Stress–Strain Behavior and Strength Characterization of Rubberized Reinforced Concrete

**DOI:** 10.3390/ma15030730

**Published:** 2022-01-19

**Authors:** Hanif Ullah, Mudassir Iqbal, Kaffayatullah Khan, Arshad Jamal, Adnan Nawaz, Nayab Khan, Fazal E. Jalal, Abdulrazak H. Almaliki, Enas E. Hussein

**Affiliations:** 1Department of Civil Engineering, University of Engineering & Technology, Peshawar 25120, Pakistan; engrhanifullah@uetpeshawar.edu.pk (H.U.); adnan.civiluet@gmail.com (A.N.); nayabkhan@uetpeshawar.edu.pk (N.K.); 2Shanghai Key Laboratory for Digital Maintenance of Buildings and Infrastructure, School of Naval Architecture, Ocean and Civil Engineering, Shanghai Jiao Tong University, Shanghai 200240, China; jalal2412@sjtu.edu.cn; 3Department of Civil & Environmental Engineering, College of Engineering, King Faisal University (KFU), Al-Ahsa 31982, Saudi Arabia; kkhan@kfu.edu.sa; 4Interdisciplinary Research Center of Smart Mobility and Logistics (IRC-SML), King Fahd University of Petroleum & Minerals, Dhahran 31261, Saudi Arabia; arshad.jamal@kfupm.edu.sa; 5Department of Civil Engineering, College of Engineering, Taif University, P.O. Box 11099, Taif 21944, Saudi Arabia; a.almaliki@tu.edu.sa; 6National Water Research Center, P.O. Box 74, Shubra El-Kheima 13411, Egypt

**Keywords:** rubberized reinforced concrete, stress–strain curve, stiffness, ductility, compression strength

## Abstract

Due to the rapid increase in population, the use of automobile vehicles increases day by day, which causes a considerable increase in the waste tires produced worldwide. Research studies are in progress to utilize scrap tires and waste rubber material in several fields to cater the pollution problems in a sustainable and environmentally friendly manner. In this research, the shredded waste tires were used in concrete to replace fine aggregates in different percentages. The fine aggregates in the rubberized concrete were replaced 10%, 15%, and 20% by rubber. The stress–strain behavior of the concrete models is then determined and compared with the already established analytical models, i.e., Modified Kent and Park Model, Mander’s model, and Razvi and Saatcioglu Model. A total of 12 standard concrete cylinders and 18 models of each type of concrete, i.e., normal concrete, reinforced rubberized concrete with 10%, 15%, and 20% addition of rubber, were fabricated. Specimens fabricated in each replacement of rubber were laterally confined, employing 3 in (76 mm) and 6 in (152 mm) c/c tie spacing. The model and cylinders were subjected to uni-axial compression tests using Universal Testing Machine (UTM). The drop in compressive strength, stress–strain constitutive law, strain limits, and overall behavior of the rubberized reinforced concrete were explored experimentally. The results were then compared with the analytical results of the established models. The research can help explore the possible future for the use of rubberized concrete for the potential application as a structural material.

## 1. Introduction

Automobile industries are increasing continuously due to rapid increase in the usage of vehicles worldwide. Many waste tires are producing day by day, which causes a considerable increase in environmental pollution and proves to be great pressure on the present solid waste management system. So, disposal of waste rubber is most necessary to minimize and reduce the associated problems globally. The disposal rate is different in different countries; the USA discard about 1.1 million tires per person per year, while Australia disposes of 48 million tires every year. About 37 million tires are produced in England, while 200,000 tons of scrap rubber are discarded in Malaysia per year. There is no reliable data available in Pakistan regarding the production and disposal rate of waster rubber [1,2,3,4]. Due to the non-decomposable and undissolvable nature of the waste rubber, there is no proper way for its disposal and hence considered the main factor for environmental pollution. Furthermore, due to the rapid depletion of dumping sites’ availability, fire potential, and health hazards, landfilling tires are not acceptable by the local authorities and the government [5,6,7]. To counter the potential hazards of rubber, one way is to use the crumb rubber in concrete production called rubberized concrete. So, this will cause a considerable increase in the economy and help reduce the environmental impacts of these wastes while saving natural resources. Rubberized concrete (RC) is a type of concrete that employs crumb rubber from already used or disposed of tires as a partial replacement of natural aggregates. Rubberized concrete has many advantages; it provides good workability, durability, and a smaller unit weight to the concrete mix and possesses better aesthetics. ASTM D6270 has mentioned the properties of shredded waste tires and can be used in civil engineering construction [8,9,10,11].

At the material level, the authors performed research using rubber as an alternative to natural aggregates in concrete and found rubberized aggregate concrete’s tensile and compressive properties. It was concluded that replacing coarse aggregates entirely with chip rubber causes a 50% decrease in tensile and 85% decrease in compressive strength of concrete. Similarly, complete replacement of fine aggregates posted a 65% decrease in compressive strength and a 50% decrease in tensile strength of concrete. Later on, it was explored that the workability of rubberized concrete is quite adequate, but its unit weight is measurably less than that of plain concrete [8,12]. Researchers worked on the thermal properties of rubberized concrete using the hotbox technique, using 5%, 10%, and 15% of scrap rubber as volume replacement of coarse aggregate. It was observed that there is no considerable change in concrete properties up to 5% replacement. However, beyond 5% substitution, the properties changed considerably, and a significant increase was also noticed in the thermal behavior [4]. In another study, the effects of harsh environments, i.e., acid, sulphate attacks, and elevated temperature on 12 different batches of concrete employing rubber crumbs. Binding materials and aggregates were replaced with ground granulated blast furnace slag and rubber, respectively. It was concluded from the tests that employing 5 to 20% scrap rubber caused a reduction in compressive strength and loss of weight due to acid and sulphate attack as compared to regular concrete [13].

In recent research work, the effects of freeze-thaw process on rubberized concrete for pavement base and sub-base materials were investigated. It was found that the resilient modulus of rubberized aggregate concrete subjected to freeze-thaw cycles was higher than that of control samples subjected to tests at 25 °C constant temperature. Therefore, this economical and viable option of blend of RCA and rubber material can be used for pavement base and sub-base material [14]. Authors recently investigated self-compacting concrete by employing 2–5 mm and 5–10 mm particle size of scrap rubber via 10%, 20%, 30%, and 40% of naturally occurring aggregate by volume. Samples of eight different mixtures were subjected to test at 7, 28, 56, and 91 days for their mechanical performance. It was concluded from the experimental results that both the compressive and tensile strength reduced with the increase of rubber amount and increased with the aging time, respectively [15].

Many research studies have been carried out in rubberized aggregate concrete production [16], where most of the research works have been carried out on the mechanical properties of rubberized concrete mixtures. However, limited studies are available on the structural behavior of reinforced rubberized concrete. To explore the effects of partial substitution of rubber in concrete on the bond slippage was experimentally undertaken in research work. Rubber percentages of 6%, 12%, 18%, and 24% as partial replacement of fine aggregate were used to fabricate test specimens. It was concluded that the bond strength of rubberized aggregate concrete was reduced by 20% compared to the reference concrete. However, the residual bond stress slightly increased by about 10% [17]. Recently, in a research work, the effects of blast loading on rubberized aggregate concrete slab were explored numerically. Rubberized aggregate concrete is protective structural material due to its energy dissipation capacity. Therefore, the numerical results of the Karagozian and Case concrete (KCC) model were compared with the experimental data. As a result, it was concluded that the resistance to blast loading of the rubberized aggregate concrete is enhanced [18].

The above discussion concludes that addition of crumb rubber replacing coarse aggregates causes a decrease in workability, flexural strength, and split tensile strength of concrete [19]. While investigating its durability, the freezing–thawing resistance and sulfate resistance was considerably enhanced [20,21,22,23,24]. It was also noticed that the ordinary concrete was dispersed under the loads while rubberized concrete had much deformation before failure. This shows increased deformability, ductility, and an enhanced energy-dissipating capacity of rubberized concrete [25,26,27]. This suggest that the rubberized concrete can be used in non-load-bearing members. The negative effect of crumb rubber on mechanical strength could be minimized and avoided by pre-treatment of the crumb rubber using modifiers [20,21,22,23,24].

Limited studies are available on the structural behavior of rubberized concrete made of replacing crumb rubber by fine aggregate. Therefore, this research investigates the effect of replacing fine aggregate by crumb rubber investigating the stress–strain behavior of columns. Different models regarding the investigation of stress–strain behavior of the confined normal concrete were employed in this research for comparison. First, Kent and Park’s model is proposed [28], which was reused in a research study to predict various parameters and their comparison [29,30,31,32]. Modified Kent and Park Model, Mander’s model [30,32,33,34], and Razvi and Saatcioglu model [35,36,37] were used to compare the experimental results of this study in accordance with previous literature.

## 2. Methodology

In this research work, standard concrete cylinders and test specimens in reduced scale were built; column specimens had cross-sectional dimensions of 6 in × 6 in (152 mm × 152 mm) and 30 in (762 mm) span length, employing rubberized reinforced concrete. Natural fine aggregates were replaced 10%, 15%, and 20% by volume via scrap rubber. The prototype of the test specimen has a column’s cross-section of 18 in × 18 in (457 mm × 457 mm). The focus of this study is to explore the stress–strain behavior of rubberized reinforced concrete. The compressive strength of the rubberized and reference concrete was experimentally explored by testing the standard concrete cylinders and stress–strain relationship of the test specimens and compared with the available models developed for normal concrete.

## 3. Experimental Program

### 3.1. Standard Concrete Cylinder and Test Specimen

A mix ratio of 1:1.80:1.60 and water to cement ratio of 0.48 were used for casting conventional concrete cylinders. In addition, rubberized concrete cylinders employing 10%, 15%, and 20% scrap rubber as partial replacement of sand by volume were also fabricated. To find the behavior and relevant parameters of reinforced concrete employing rubber crumbs, three classes: A, B, and C, were selected for casting test specimens. Table 1 shows the specification of the test specimens.

### 3.2. Materials

Ordinary Portland cement, reinforcement, aggregates, crumb rubber, and mixing water were used in this research work. Crumb rubber was used in the form of dust obtained from retaining on sieve No. 200. The sieve analysis of coarse and fine aggregate is shown in Table 2 and Table 3, respectively. Deformed bar conforming to A615 (G 40) was used as main and longitudinal reinforcement in the test specimen; #1 was provided as shear reinforcement having 3 in (76 mm) and 6 in (152 mm) center-to-center spacing, while 8#2 were provided as main longitudinal reinforcement. Ties spacing of 3 in (76 mm) c/c in the specimens 10RRC3, 15RRC3, and 20RRC3 and 6 in (152 mm) c/c in the specimens 10RRC6, 15RRC6, and 20RRC6 were provided. The formwork and reinforcement used during specimen fabrication are shown in Figure 1c.

### 3.3. Fabrication Phase

First, the reference concrete cylinders were fabricated. Then, 10%, 15%, and 20% rubberized concrete cylinders and specimens were fabricated in three consecutive days and were placed in the water tank for 28 days of curing. For proper concrete placement, a tamping rod made of steel was used while concreting in standard concrete cylinders. However, a mechanical vibrator of a 1-inch diameter was used to remove voids and entrapped air during the fabrication of the test specimens.

### 3.4. Test Specimens

For observance and clarity of cracks during the tests, specimens were whitewashed. The concrete cylinders and specimens were then labelled as shown in Figure 1d.

A total of 12 standard concrete cylinders and eighteen test specimens were constructed for the evaluation of mechanical properties. Figure 1a shows the long and cross-section of the specimen. The specimen nomenclature is given such that first two digits shows percent replaced fine aggregate by crumb rubber, RRC denotes Rubberized Reinforced Concrete and the last digit reflect the spacing of ties in inches. For instance, column specimen 10RRC6 means RRC with 10% fine aggregate replacement by crumb rubber with ties spacing of 6 inches c/c.

### 3.5. Experimental Investigation

The experimental investigation of the models revealed the stress–strain curve of reduced scale specimens built in reinforced concrete employing rubber crumb. The curves were compared with the previously determined models for normal reinforced concrete having the same mix ratio of 1:1.80:1.60 with a water to cement ratio of 0.48. The strength of crumb rubber concrete is determined experimentally by testing the specimens with partial replacement of fine aggregates via rubber. Variation in the behavior due to lateral ties’ spacing was also explored in this research by employing 3 in (76 mm) and 6 in (152 mm) spacing of transverse reinforcement.

### 3.6. Testing Setup

The test specimens were examined experimentally under uni-axial load (monotonic 2 k/in^2^/min) using the Universal Testing Machine (UTM) available at the Structural Engineering Laboratory, Civil Engineering Department, University of Engineering and Technology Peshawar, Khyber Pakhtunkhwa, Pakistan. Two linear variable displacement transducers (LVDTs) were placed on opposite sides of the specimen to measure average axial deformation. Test data were recorded and transferred to the computer accordingly. The UTM and LVDT utilized in this research study can be seen in Figure 1b.

## 4. Results and Discussion

Data recorded during the compression test of cylinders and models were transferred into the computer for analysis. In addition, the results obtained for rubberized reinforced concrete were compared with the available models for reference concrete of the same mix ratio.

### 4.1. Observed Behavior and Comparison of 10RRC6 Models

It was observed in the specimens of rubberized reinforced concrete having 10% addition of rubber and tie bar having 6 in spacing, that the increase in load caused an increase in vertical stresses, followed by failure and spalling of the cover concrete. Further increase in load caused the failure of core concrete and then buckling of the longitudinal reinforcement while the tie bar remained unopened in all three specimens. The failure pattern is shown in Figure 2a.

The trend of the test specimens’ results was contrasted with the Modified Kent and Park Model of corresponding normal concrete in Figure 2. It can be observed that the strain at peak and peak strength of rubberized concrete specimens increased. However, the relative ductility, stiffness, and ultimate strain of rubberized concrete decreased in comparison with the Modified Kent and Park Model. Figure 2b illustrates the graphical comparison of the test specimens’ results with the model. The test results of the specimens were then compared with the analytical results of Mander’s Model. It can be noted that the strain at peak, peak strength, and ultimate strain increased while the stiffness and relative ductility decreases. Figure 2c indicates the graphical comparison of the test specimens’ results with Mander’s Model.

Similarly, the trend of the results of the specimens was finally compared with the analytical values of the Razvi and Saatcioglu Model. Table 4 shows that the strain at peak and peak strength increased while the relative ductility, ultimate strain, and stiffness decreased. Figure 2d shows the graphical variation and comparison of the test specimens’ results with the model.

### 4.2. Observed Behavior and Comparison of 10RRC3 Models

It was noticed in the specimens of rubberized reinforced concrete having 10% addition of rubber and tie bar employing 3 in spacing that the increase in load caused an increase in vertical stresses, followed by failure and spalling of the cover concrete. Further increase in load caused failure of the core concrete followed by a buckling of the longitudinal reinforcement while the tie bar remained unopened in all three specimens of this type. The failure pattern is shown in Figure 3a. The experimental values found from the tests of the specimens given in Table 5 were matched with the analytical values of the Modified Kent and Park Model. It can be concluded that the strain at peak and peak strength increased while the stiffness, relative ductility, and ultimate strain decreased. Figure 3b illustrates the comparison of the test specimens’ results graphically with the model. The experimental findings were also matched with Mander’s model. It was concluded that the strain at peak and peak strength increased while the stiffness, ultimate strain, and relative ductility decreased. Figure 3c graphically compares the test specimens’ results with the Mander’s Model.

Finally, the results were compared with the third model, i.e., Razvi and Saatcioglu Model. The strain at peak and peak strength increased while the relative ductility, stiffness, and ultimate strain decreased. Graphical comparison and variation of the test specimens results are shown in Figure 3d.

#### Comparison of 10RRC6 and 10RRC3 Specimens

Table 6 shows the average experimental values of 10RRC6 and 10RRC3 specimens. It is evident that the stiffness and peak strength of 10RRC3 increased by 16.62% and 7.9% in contrast to 10RRC6, respectively, due to decreasing the spacing of the ties. Similarly, the strain at peak and ultimate strain of 10RRC3 has increased by 21.62% and 25.86%, respectively. However, a reduction of 3.8% was found in the relative ductility of 10RRC3 compared to 10RRC6 due to the higher stiffness of the 10RRC3.

### 4.3. Observed Behavior and Comparison of 15RRC6 Models

It was observed in the specimens of rubberized reinforced concrete having 15% addition of rubber and tie bar of spacing 6 in (152 mm) that the increase in load caused an increase in vertical stresses, followed by failure and spalling of the cover concrete. Further increase in load caused the failure of core concrete followed by a buckling of the longitudinal reinforcement while the tie bar remained unopened in all three specimens of this type. The failure pattern is shown in Figure 4a. The average results of 15RRC6 specimens were compared with the analytical model, i.e., Modified Kent and Park Model. Figure 4 shows that the strain at peak and peak strength increased while the relative ductility, stiffness, and ultimate strain decreased. Graphical variation and comparison of the test specimens’ results can be seen in Figure 4b. Likewise, the results were compared with the second model of corresponding normal concrete, i.e., Mander’s Model. Table 7 shows that the strain at peak and peak strength increased while the stiffness, ultimate strain, and relative ductility decreased. Graphical variation and comparison of the test specimens’ results with the model are shown in Figure 4c.

Similarly, test results of 15RRC6 specimens and those of the analytical model, given in Table 7, were also compared. Again, strain at peak and peak strength increased while the relative ductility, stiffness, and ultimate strain decreased compared to the Razvi and Saatcioglu Model. Figure 4d shows the graphical comparison of the test specimens’ results Model.

### 4.4. Observed Behavior and Comparison of 15RRC3 Models

It was found in the specimens of rubberized reinforced concrete having 15% of rubber and tie bar of spacing 3 in (76 mm), that the increase in load causes an increase in vertical stresses, followed by failure and spalling of the cover concrete. Further increase in load causes the failure of core concrete followed by a buckling of the longitudinal reinforcement while the tie bar remained unopened in all three specimens of this type. The failure pattern is shown in Figure 5a. The experimental results of the specimens and the analytical model results are shown in Figure 5. The strain at peak and peak strength increased while the relative ductility, stiffness, and ultimate strain decreased compared to the corresponding normal concrete model, i.e., Modified Kent and Park Model. Results of the specimens and their comparison with the analytical model are graphically shown in Figure 5b.

The specimens’ results were also compared with the second model, i.e., Mander’s Model for normal concrete. Strain at peak and peak strength increased while the relative ductility, stiffness, and ultimate strain decrease. Figure 5c shows the graphical comparison of the test specimens’ results with the model. Likewise, the results of the test specimens were matched with the 3rd model, i.e., Razvi and Saatcioglu Model. It can be seen from Table 8 that the strain at peak and peak strength increased while the relative ductility, stiffness, and ultimate strain decrease. Figure 5d shows the graphical comparison of the test specimens’ results.

#### Comparison of the Behavior of 15RRC6 and 15RCC3

Experimental results from the tests of both types of specimens, i.e., 15RRC6 and 15RRC3 were also compared in Table 9. It is evident from the average values that the stiffness and peak strength of 15RRC3 has increased by 5.74% and 3.2%, respectively, as compared to 15RRC6. Furthermore, the strain at peak and ultimate strain of 15RRC3 has increased by 13.51% and 17.24%, respectively.

### 4.5. Observed Behavior and Comparison of 20RRC6 Models

It was noticed in the specimens of rubberized reinforced concrete having 20% of rubber and tie bar of spacing 6 in (152 mm) that the increase in load causes an increase in vertical stresses, followed by failure and spalling of the cover concrete. Further increase in load causes the failure of core concrete followed by a buckling of the longitudinal reinforcement while the tie bar remained unopened in all three specimens of this type. The failure pattern is shown in Figure 6a. The average results obtained from the tests of the specimens were matched with the results of the analytical model, i.e., Modified Kent and Park Model. Figure 6 indicates that the strain at peak, peak strength increased while the relative ductility, stiffness, and ultimate strain decreases. Figure 6b shows the graphical comparison of the test specimens’ results with that of the model.

The results were also compared with the second model established for reference concrete. It can be seen from Figure 6 that the strain at peak and peak strength increased while the relative ductility, stiffness, and ultimate strain decreases in contrast to the model, i.e., Mander’s Model. Figure 6c shows the graphical comparison of the test specimens’ results with the model. Likewise, the results of the specimens were also compared with those of the established model, i.e., Razvi and Saatcioglu Model. It can be seen from Table 10 that the strain at peak and peak strength increased while the relative ductility, stiffness, and ultimate strain decreasd in contrast to the model’s results. Figure 6d shows the graphical comparison of the test specimens’ results with the model.

### 4.6. Observed Behavior and Comparison of 20RRC3 Models

It was found in the specimens of rubberized reinforced concrete having 20% of rubber and tie bar of spacing 3 in (76 mm), that the increase in load causes an increase in vertical stresses, followed by failure and spalling of the cover concrete. Further increase in load causes the failure of core concrete followed by a buckling of the longitudinal reinforcement while the tie bar remained unopened in all three specimens of this type. The failure pattern is shown in Figure 7a.

At last, the test specimens’ behavior and parameters were compared with the model i.e., Modified Kent and Park Model. It can be observed from Figure 7 that the strain at peak and peak strength increased while the relative ductility, stiffness, and ultimate strain decreased in contrast to the model’s parameters. Figure 7b illustrates the variation and comparison of the test results with that of the model. The obtained results of the test specimens were also compared with Mander’s Model. Strain at peak and peak strength increased while the relative ductility, stiffness, and ultimate strain decrease. Figure 7c shows the graphical comparison of the test specimens’ results with the model. Likewise, the results were then compared with the 3rd model, i.e., Razvi and Saatcioglu Model. Table 11 shows that the strain at peak and peak strength increased while the relative ductility, stiffness, and ultimate strain decreased compared to the analytical model’s results. Figure 7d shows the graphical comparison of the test specimens’ results with the model.

#### Comparison of 20RRC6 and 20RRC3

Various parameters of 20RRC6 specimens were compared in Table 12 with that of 20RRC3. It is shown in that the stiffness almost remained the same for both types of specimens. However, the peak strength of 20RRC3 is increased by 8.6% as compared to 20RRC6. The strain at peak and ultimate strain of 20RRC3 has increased by 28.57% and 12.3%, respectively, respectively, in contrast to 20RRC6. Relative ductility of 20RRC3 was reduced by 12.2% in contrast to 20RRC6 specimens.

### 4.7. Stiffness Degradation

The stiffness of the models decreases by increasing the percentage of rubber. The decrease in stiffness of the members also depends upon the spacing of the lateral ties. However, the degradation does not follow a specific trend. The average stiffness of each percentage can be seen Figure 8a,b for 6 in (152 mm) and 3 in (76 mm) c/c tie spacing, respectively.

### 4.8. Reduction in Weight

Rubber being a lightweight material as compared to sand; significant reduction in the weight of concrete was recorded due to partial replacement of sand via different percentages of rubber by volume. The effect of rubber addition on weight can be seen in Figure 8c.

### 4.9. Relative Ductility

It was found from the test data that the relative ductility decreases due to the incorporation of rubber in concrete as compared to corresponding normal concrete. It was explored that the decrease in relative ductility of the members also depends upon the spacing of the lateral ties. However, the degradation does not follow a specific trend. The relative ductility of each percentage can be seen in Figure 8d,e for 6 in (152 mm) and 3 in (76 mm) c/c tie spacing, respectively.

### 4.10. Compressive Strength

Due to less adhesion between the rubber and the cement matrix, the compressive strength generally decreases by increasing the addition of rubber. The compressive strength of all types of specimens can be seen in Figure 8f.

## 5. Conclusions

This research study focused on exploring the stress–strain relationship of rubberized reinforced concrete test column specimens (scale down) and their comparison with the established analytical models. The following conclusions have been summarized from the current research work.
The experimental data recorded during testing of the concrete cylinders and reinforced test specimens shows that 20% rubber replacement via fine aggregates causes an 8.4% reduction in weight and 41.86% in compressive strength compared to normal concrete. The peak strength has increased from 7% to 14% concerning the Modified Kent and Park Model. The reinforced rubberized concrete stiffness decreased while the strain at peak increased.Likewise, the ultimate strain was decreased in each series, and the degradation of stiffness occurs at 50.53%, 64.72%, and 74.34% concerning normal concrete. Similarly, employing scrap rubber crumbs in concrete causes the relative ductility to decrease by 39.60%, 44.93%, and 51.27% compared with Modified Kent and Park Model, for 10%, 15%, and 20% volume replacement (3 inches spacing). It was also found that the relative ductility is reduced for specimens having 3 in (76 mm) c/c tie spacing in contrast to specimens having 6 in (152 mm) c/c tie spacing by using rubberized concrete due to confinement of lateral ties.The plain and reinforced concrete employing crumb rubber is under investigation worldwide at different sections and full-frame structure levels. The Poison’s ratio of rubberized aggregate concrete and its variation with percentage replacement of rubber is under exploration. It is recommended to build frame structures and various sections of structural members, i.e., beam and column of plain and reinforced concrete, and be tested under static and dynamic loading to know the performance parameters and behavior so that scrap rubber can be confidently used as building materials. This will serve as a guideline of reusability of rubber in construction industries and will minimize the associated problems like contamination of the environment and dumping site availability.With the advancement in artificial intelligence (AI), wide variety of civil engineering problems are solved using AI [38,39,40,41,42,43,44,45]. AI models shall be developed that can product the mechanical and durability properties of rubberized concrete. This way, cost, time, and economy of the project can be effectively increased.


## Figures and Tables

**Figure 1 materials-15-00730-f001:**
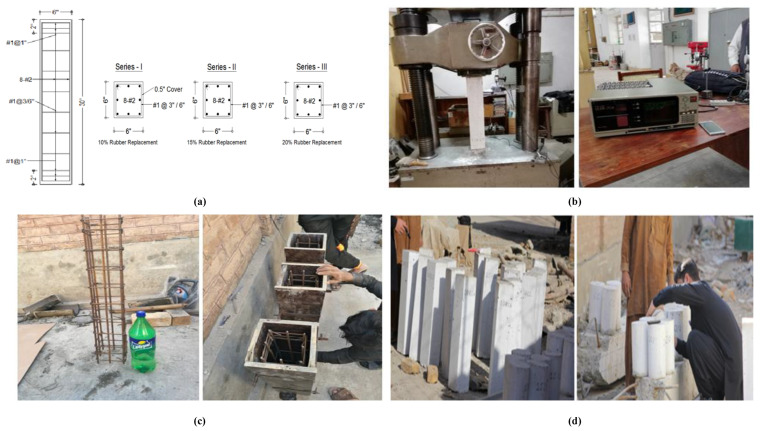
Models fabrication phase: (**a**) long and cross-section of the specimen, (**b**) universal testing machine, LVDT, and data logger, (**c**) reinforcement and formwork of the specimen, (**d**) fabricated and labeled standard concrete cylinders and test specimens.

**Figure 2 materials-15-00730-f002:**
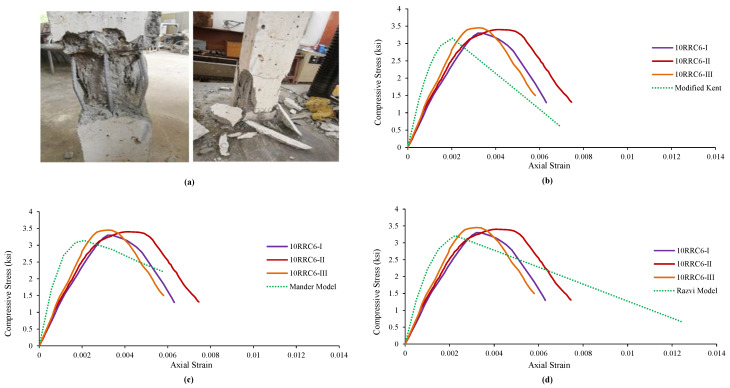
Damages and Ơ–Ɛ curves of 10RRC6 and their comparison with the models of reference concrete: (**a**) Damages and behavior of 10RRC6 specimen, (**b**) Comparison with Modified Kent Model, (**c**) Comparison with Mander Model, (**d**) Comparison with Razvi Model.

**Figure 3 materials-15-00730-f003:**
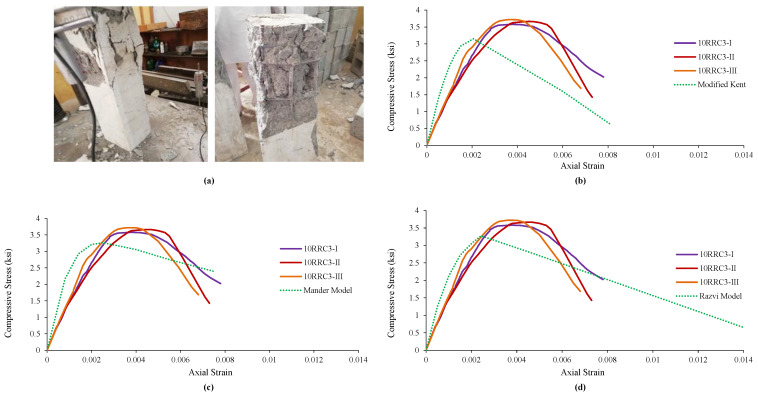
Damages and Ơ–Ɛ curves of 10RRC3 along with its comparison with various analytical models of reference concrete: (**a**) Damages and behavior of 10RRC3 specimen, (**b**) comparison with Modified Kent Model, (**c**) comparison with Mander’s Model, (**d**) comparison with Razvi Model.

**Figure 4 materials-15-00730-f004:**
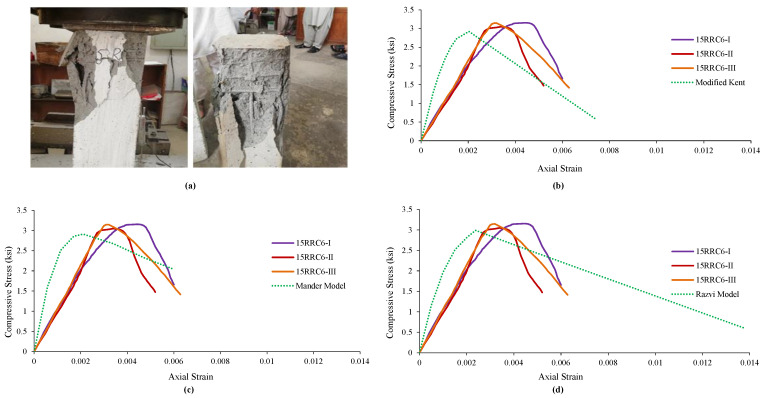
Damages and Ơ–Ɛ curves of 15RRC6 along with its comparison with various analytical models of reference concrete: (**a**) Damages and behavior of 15RRC6 specimen, (**b**) comparison with Modified Kent Model, (**c**) comparison with Mander Model, (**d**) comparison with Razvi Model.

**Figure 5 materials-15-00730-f005:**
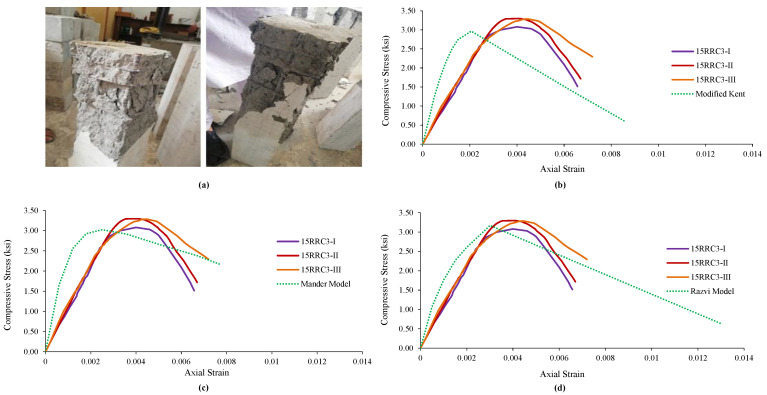
Damages and Ơ–Ɛ curves of 15RRC3 along with its comparison with various analytical models of reference concrete: (**a**) Damages and behavior of 15RRC3 specimen, (**b**) Comparison with Modified Kent Model, (**c**) Comparison with Mander Model, (**d**) Comparison with Razvi Model.

**Figure 6 materials-15-00730-f006:**
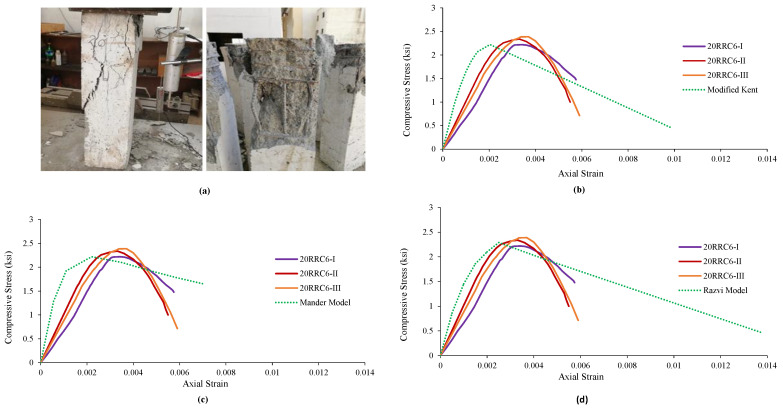
Damages and Ơ–Ɛ curves of 20RRC6 along with its comparison with various analytical models of reference concrete: (**a**) Damages and behavior of 20RRC6 specimen, (**b**) comparison with Modified Kent Model, (**c**) comparison with Mander Model, (**d**) comparison with Razvi Model.

**Figure 7 materials-15-00730-f007:**
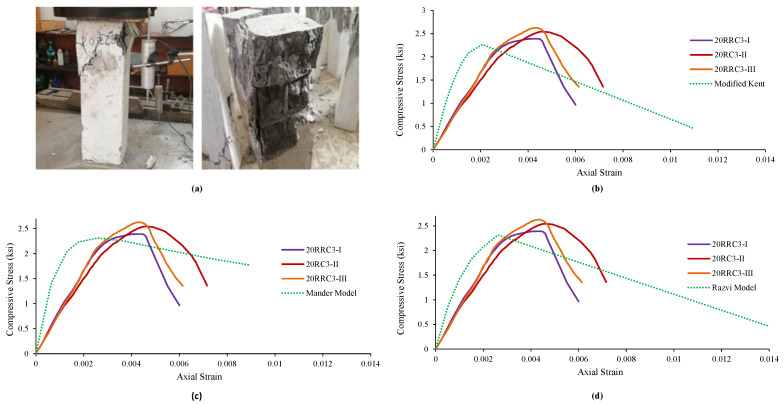
Damages and Ơ–Ɛ curves of 20RRC3 along with its comparison with various analytical models of reference concrete: (**a**) Damages and behavior of 20RRC3 specimen, (**b**) comparison with Modified Kent Model, (**c**) comparison with Mander’s Model, (**d**) comparison with Razvi Model.

**Figure 8 materials-15-00730-f008:**
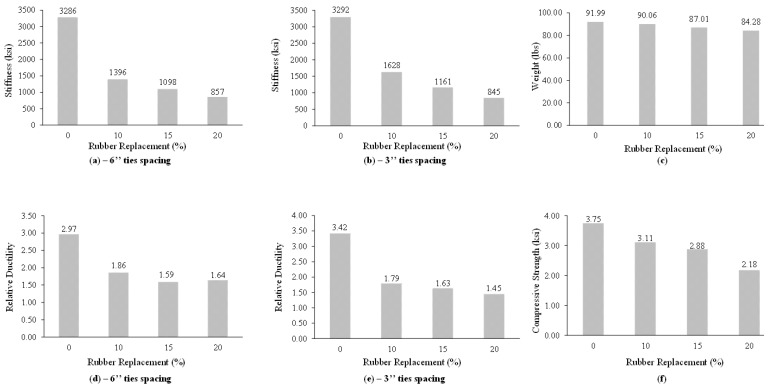
Effect of percentage replacement of scrap rubber on the properties of rubberized concrete: (**a**,**b**) Stiffness of specimen with 6” and 3” stirrups spacing, (**d**,**e**) Ductility of the specimen with 6” and 3” stirrups spacing, (**c**) Weight variation and (**f**) Compressive strength variation.

**Table 1 materials-15-00730-t001:** Specimen details.

Class	Specimen ID	Ties Spacing	Reinforcement (Longitudinal)	Rubber (%) by Volume	Cross-Section (in^2^)
A1	10RRC6	#1@ 6 in c/c	8 # 2	10	6 × 6
A2	10RRC3	#1@ 3 in c/c			
B1	15RRC6	#1@ 6 in c/c		15	6 × 6
B2	15RRC3	#1@ 3 in c/c			
C1	20RRC6	#1@ 6 in c/c		20	6 × 6
C2	20RRC3	#1@ 3 in c/c			

Note: RRC stands for Rubberized Reinforced Concrete.

**Table 2 materials-15-00730-t002:** Sieve Analysis of Coarse Aggregates.

Sieve No	Sieve Size (mm)	Retained Weight (gm)	Passing Weight (gm)	Cumulative Weight (gm)	%Age Passing
3/4”	19	0	3998	0	100
1/2”	12.5	61	3937	61	98.5
3/8”	9.5	1242	2695	1303	67.4
4	4.75	1708	987	3011	24.7
Pan		987	0	3998	0.0

**Table 3 materials-15-00730-t003:** Sieve Analysis of Fine Aggregates.

Sieve No	Sieve Size (mm)	Retained Weight (gm)	Passing Weight (gm)	Cumulative Weight (gm)	%Age Passing
4	4.75	0	1008	0	100.0
8	2.35	0	1008	0	100.0
16	1.18	1	1007	1	99.9
30	0.6	4	1003	5	99.5
50	0.3	34	969	39	96.1
100	0.15	856	113	895	11.2
200	0.075	103	10	998	1.0
Pan		10	0	1008	0.0

**Table 4 materials-15-00730-t004:** Results of 10RRC6 specimens and their comparison with the analytical models.

Parameter	10RRC6 (Average)	MKM	% Difference w.r.t MKM	MM	% Difference w.r.t MM	RM	% Difference w.r.t RM
Peak Strength (ksi)	3.38	3.15	(+) 07.30	3.14	(+) 07.64	3.19	(+) 05.95
Ultimate Strain (in/in)	0.0065	0.0069	(−) 05.79	0.0058	(+) 12.06	0.0125	(−) 48.00
Strain at Peak (in/in)	0.0035	0.002	(+) 75.00	0.0021	(+) 66.66	0.0022	(+) 59.09
Relative Ductility	1.86	3.4	(−) 45.29	2.76	(−) 32.60	5.58	(−) 66.66
Stiffness (ksi)	1396	3108	(−) 55.08	3139	(−) 55.52	3160	(−) 55.82

Where MKM = Modified Kent Model, MM = Mander Model and RM = Razvi Model.

**Table 5 materials-15-00730-t005:** Results of 10RRC3 specimens and their comparison with the analytical models.

Parameter.	10RRC3 (Average)	MKM	% Difference w.r.t MKM	MM	% Difference w.r.t MM	RM	% Difference w.r.t RM
Peak Strength (ksi)	3.65	3.2	(+) 14.06	3.27	(+) 11.62	3.24	(+) 12.65
Ultimate Strain (in/in)	0.0073	0.0081	(−) 09.87	0.0075	(−) 02.14	0.014	(−) 47.85
Strain at Peak (in/in)	0.0045	0.0021	(+) 114.28	0.0025	(+) 80.00	0.002	(+) 87.50
Relative Ductility	1.79	3.94	(−) 54.56	2.98	(−) 39.93	5.76	(−) 68.92
Stiffness (ksi)	1628	3292	(−) 50.54	3150	(−) 48.31	3171	(−) 48.65

Where MKM = Modified Kent Model, MM = Mander Model and RM = Razvi Model.

**Table 6 materials-15-00730-t006:** Results of 10RRC3 and 10RRC6 specimens.

Parameter	10RRC6 Average	10RRC3 Average	% Difference w.r.t 10RRC3
Peak Strength (ksi)	3.38	3.65	(+) 07.39
Ultimate Strain (in/in)	0.0058	0.0073	(+) 20.54
Strain at Peak (in/in)	0.0037	0.0045	(+) 17.77
Relative Ductility	1.86	1.79	(−) 03.91
Stiffness (ksi)	1396	1628	(+) 14.25

**Table 7 materials-15-00730-t007:** Results of 15RRC6 specimens and their comparison with the analytical models.

Parameter	15RRC6 (Average)	MKM	% Variation w.r.t MKM	MM	% Variation w.r.t MM	RM	% Variation w.r.t RM
Peak Strength (ksi)	3.12	2.92	(+) 06.84	2.92	(+) 04.45	2.92	(+) 07.87
Ultimate Strain (in/in)	0.0058	0.0074	(−) 21.62	0.0074	(−) 29.72	0.0074	(−) 14.86
Strain at Peak (in/in)	0.0037	0.002	(+) 85.00	0.002	(+) 70.00	0.002	(+) 60.00
Relative Ductility	1.59	3.65	(−) 56.43	3.65	(−) 58.08	3.65	(−) 46.30
Stiffness (ksi)	1098	2874	(−) 61.79	2874	(−) 64.40	2874	(−) 61.16

Where MKM = Modified Kent Model, MM = Mander Model and RM = Razvi Model.

**Table 8 materials-15-00730-t008:** Results of 15RRC3 specimens and their comparison with the analytical models.

Parameter	15RRC3 (Average)	MKM	% Variation w.r.t MKM	MM	% Variation w.r.t MM	RM	% Variation w.r.t RM
Peak Strength (ksi)	3.22	2.96	(+) 08.78	3.02	(+) 06.62	3.17	(+) 01.57
Ultimate Strain (in/in)	0.0068	0.0086	(−) 20.93	0.0074	(−) 08.10	0.013	(−) 47.69
Strain at Peak (in/in)	0.0042	0.0021	(+) 100.00	0.0025	(+) 68.00	0.003	(+) 40.00
Relative Ductility	1.63	4.18	(−) 61.00	2.96	(−) 44.93	4.32	(−) 62.26
Stiffness (ksi)	1161	2877	(−) 59.64	3023	(−) 61.59	2891	(−) 59.84

Where MKM = Modified Kent Model, MM = Mander Model and RM = Razvi Model.

**Table 9 materials-15-00730-t009:** Results of 15RC6 and 15RCC3 specimens.

Parameter	15RRC6 (Average)	15RRC3 (Average)	% Variation w.r.t 15RRC3
Peak Strength (ksi)	3.12	3.22	(+) 03.10
Ultimate Strain (in/in)	0.0058	0.0068	(+) 14.70
Strain at Peak (in/in)	0.0037	0.0042	(+) 11.90
Relative Ductility	1.59	1.63	(+) 02.45
Stiffness (ksi)	1098	1161	(+) 05.42

**Table 10 materials-15-00730-t010:** Results of 20RRC6 specimens and their comparison with the analytical models.

Parameter	20RRC6 (Average)	MKM	% Variation w.r.t MKM	MM	% Variation w.r.t MM	RM	% Variation w.r.t RM
Peak Strength (ksi)	2.32	2.22	(+) 04.50	2.22	(+) 04.50	2.29	(+) 01.31
Ultimate Strain (in/in)	0.0057	0.0099	(−) 42.42	0.007	(−) 18.57	0.0138	(−) 58.69
Strain at Peak (in/in)	0.0035	0.002	(+) 75.00	0.0022	(+) 59.09	0.0025	(+) 40.00
Relative Ductility	1.64	4.86	(−) 66.25	3.18	(−) 48.42	5.52	(−) 70.28
Stiffness (ksi)	857	2171	(−) 60.25	2612	(−) 67.18	2234	(−) 61.63

Where MKM = Modified Kent Model, MM = Mander Model and RM = Razvi Model.

**Table 11 materials-15-00730-t011:** Results of 20RRC3 specimens and their comparison with the analytical models.

Parameter	20RRC3 (Average)	MKM	% Variation w.r.t MKM	MM	% Variation w.r.t MM	RM	% Variation w.r.t RM
Peak Strength (ksi)	2.52	2.26	(+) 11.50	2.31	(+) 09.09	2.31	(+) 09.09
Ultimate Strain (in/in)	0.0064	0.011	(−) 41.81	0.00889	(−) 28.00	0.014	(−) 54.28
Strain at Peak (in/in)	0.0045	0.0021	(+) 114.28	0.0026	(+) 73.07	0.0026	(+) 73.07
Relative Ductility	1.44	5.29	(−) 72.77	3.42	(−) 57.89	5.38	(−) 73.23
Stiffness (ksi)	845	2171	(−) 61.07	2566	(−) 67.06	2261	(−) 62.54

Where, MKM = Modified Kent Model, MM = Mander Model and RM = Razvi Model.

**Table 12 materials-15-00730-t012:** Results of 20RRC6 and 20RRC3 specimens.

Parameter	20RRC6 (Average)	20RRC3 (Average)	% Variation w.r.t 20RRC3
Peak Strength (ksi)	2.32	2.52	(+) 07.93
Ultimate Strain (in/in)	0.0057	0.0064	(+) 10.93
Strain at Peak (in/in)	0.0035	0.0045	(+) 22.22
Relative Ductility	1.64	1.44	(−) 13.88
Stiffness (ksi)	857	845	(−) 01.42

## Data Availability

All data used in the manuscript has been reported in the main text.

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
