# Peer review of "Experimental Investigation of the Stress–Strain Behavior and Strength Characterization of Rubberized Reinforced Concrete"

_materials, 2022, doi:10.3390/ma15030730_

Round 1

Reviewer 1 Report

I have carefully read the manuscript entitled “Experimental investigation of the stress-strain behavior and strength characterization of rubberized reinforced concrete” and analyzed its potential for publication in the MDPI journal Materials (ISSN 1996-1944).

In my opinion, the manuscript is interesting and involves recent topics. The utilization of used tires in the construction industry is widely discussed, so I assume that the article will find rather wide audience. The manuscript is fairly written and I suggest accepting it for publication after minor revision.

Below I have listed some weaknesses of the manuscript:

  1. The introduction is too long and touches too many issues. I suggest writing a new shorter Introduction that will point out the novelty and scientific value of presented work on the canvas of contemporary literature. At this point, I would like to say that the Introduction the authors have prepared is really interesting and I encourage them to extent it and submit as a review article.
  2. The manuscript is lacking a description of the aim of the study. This should be added.
  3. The list of materials used is a little too simplified. Please provide some more information identifying the aggregates and crumb rubber.
  4. In lines 220, 230, 240, 244 and many more after that I have found the citation error message instead of reference to something important. Please correct.
  5. In the Table 1 it should be indicated that the % is volumetric.
  6. In Figures 2, 3, 4, 5, 6, and 7 the captions contain unknown symbols that were not defined anywhere in the text. Please define the symbols or change captions.
  7. The conclusions contain a mere list of some observations. In the scientific paper I would rather expect some explanation not only a list of facts.

Reviewer 2 Report

In this article, the authors have investigated the stress-strain behavior and strength characterization of rubberized reinforced concrete. The detailed comments are as follows:

  1. The abstract is well-written.
  2. Lines 45-47 are unnecessary. The article is targeting a global community.
  3. The study discussed in lines 66 to 71 is not purely related to concrete. Therefore, it should be removed from the article. Authors may discuss some latest work on rubberized concrete. For example https://doi.org/10.1016/j.resconrec.2020.105353
  4. The introduction section needs revision. It is too long. Authors should only discuss the works related to their study. Authors may consider the above-mentioned reference to improve the introduction section.
  5. There are a few errors in citations of articles.
  6. The loading rate is missing.
  7. Line 516, it should be promising instead of compromising.
  8. Section 6 is too general and does not relate to the findings of this study. It should be removed.
  9. The authors have only compared the results with other available models. It is better that authors should develop their own model.
  10. Authors have compared their results with models that are not developed for rubberized concrete. Therefore, it is obvious that those models are not effective in predicting the performance of concrete. Authors may compare the results of this study with other studies to add some innovation in this study.

Round 2

Reviewer 2 Report

The authors have revised the article as per the reviewer's comments. Therefore, it is suitable for publication.